# Does membrane feeding compromise the quality of *Aedes aegypti* mosquitoes?

**Perran A. Ross**  *, **Meng-Jia Lau, Ary A. Hoffmann**

Pest and Environmental Adaptation Research Group, Bio21 Institute and the School of BioSciences, The University of Melbourne, Parkville, Victoria, Australia

* perran.ross@unimelb.edu.au

**Data Availability Statement:** All relevant data are within the manuscript and its Supporting Information files.

**Funding:** AAH was supported by the National Health and Medical Research Council (1132412,

## Abstract

Modified *Aedes aegypti* mosquitoes are being mass-reared for release in disease control programs around the world. Releases involving female mosquitoes rely on them being able to seek and feed on human hosts. To facilitate the mass-production of mosquitoes for releases, females are often provided blood through artificial membrane feeders. When reared across generations there is a risk that mosquitoes will adapt to feeding on membranes and lose their ability to feed on human hosts. To test adaptation to membrane feeding, we selected replicate populations of *Ae. aegypti* for feeding on either human arms or membrane feeders for at least 8 generations. Membrane-selected populations suffered fitness costs, likely due to inbreeding depression arising from bottlenecks. Membrane-selected females had higher feeding rates on membranes than human-selected ones, suggesting adaptation to membrane feeding, but they maintained their attraction to host cues and feeding ability on humans despite a lack of selection for these traits. Host-seeking ability in small laboratory cages did not differ between populations selected on the two blood sources, but membrane-selected females were compromised in a semi-field enclosure where host-seeking was tested over a longer distance. Our findings suggest that *Ae. aegypti* may adapt to feeding on blood provided artificially, but this will not substantially compromise field performance or affect experimental assessments of mosquito fitness. However, large population sizes (thousands of individuals) during mass rearing with membrane feeders should be maintained to avoid bottlenecks which lead to inbreeding depression.

## Introduction

One of the most effective ways to reduce pathogen transmission by mosquitoes is to suppress their populations, but traditional approaches are becoming increasingly ineffective. *Aedes aegypti*, the principal vector of dengue, is becoming resistant to insecticides that are widely used to control virus transmission [1]. Modified mosquitoes are now being mass-reared and released into the field as an alternative way of reducing pathogen transmission. Some programs involve male-only releases that aim to suppress populations. Males that are irradiated [2], genetically modified [3] or infected with *Wolbachia* [4] can be released into the field and mate with wild females, reducing their fertility. Other programs involve releases of both males

1118640, www.nhmrc.gov.au). The funders had no role in study design, data collection and analysis, decision to publish, or preparation of the manuscript.

**Competing interests:** The authors have declared that no competing interests exist.

and females which aim to replace natural populations with mosquitoes that have a reduced capacity to transmit viruses. *Aedes* mosquitoes with *Wolbachia* infections that block dengue transmission have now been released in several countries [5], while gene drives have recently been developed in *Anopheles* mosquitoes for both population replacement [6, 7] and suppression [8].

For male-only releases, the ability of released males to seek and mate with wild females is critical [9], while for releases involving both sexes, females must be able to survive and reproduce in natural populations. Mosquito stocks that are mass-reared for release are typically maintained under controlled conditions in a laboratory [10]. Insects can adapt to captivity, leading to reduced fitness under natural conditions (reviewed in [11]). This can be an issue when colonies are reared in the laboratory across generations and then released in biological control programs. In *Ae. aegypti* mosquitoes, laboratory adaptation can affect host-seeking behavior [12] and reduce levels of insecticide resistance [13] which may influence the success of both population suppression and population replacement programs. Laboratory rearing can also reduce female fertility and male mating success through inbreeding depression and drift if population sizes are not maintained at sufficiently high levels [14].

Domestic *Ae. aegypti* are anautogenic, and rearing them in the laboratory requires a source of blood for egg production. Females can be fed blood from a variety of vertebrate hosts including guinea pigs [15], sheep [10] and humans [16]. *Aedes aegypti* are highly anthropophilic [17] and have greater fertility when fed human blood compared to non-human blood [18], particularly when infected with *Wolbachia* symbionts [19–21]. However, non-human blood is frequently used during laboratory rearing as often it is more easily obtained, is subject to fewer regulations and poses lower risks of virus transmission. Artificial diets have also been developed that may be suitable alternatives to blood for mass-rearing mosquitoes [22–25].

In the laboratory, blood is provided to mosquitoes in two main ways: either directly from a live animal or through an artificial membrane feeder. Membrane feeders are often used when rearing mosquitoes on a large scale, where there are concerns with animal welfare or where it is not feasible to use human volunteers for ethical or practical reasons [26]. Several membrane feeding devices have been developed, including commercial products and in-house designs constructed from basic materials [27]. Membrane feeders typically consist of a reservoir containing the blood (or artificial diet) and a membrane, usually collagen or Parafilm, through which mosquitoes can access the blood. The blood can be pre-heated, usually to 37–40°C, or warmed through a heating element in the feeder. Most studies comparing membrane feeders to live hosts report similar feeding proportions [28–31] and female fertility [32, 33] between the two sources, but designs and membrane materials can differ in their efficacy [34–36].

While maintaining mosquitoes in the laboratory for experiments or field releases, there is potential for adaptation to membrane feeding. Deng et al. [32] observed that *Aedes albopictus* females had lower feeding rates on membrane feeders than on a live guinea pig, but after three generations of selection on each blood source they exhibited similar feeding rates. Membrane feeders may impose different selection pressures to live hosts when maintaining mosquitoes in the laboratory. Mosquitoes use a combination of heat, odor, $CO_2$ and visual cues to locate hosts [37–40], but most cues are absent from membrane feeders. Mosquitoes feeding on a live host must pierce the skin and probe for a blood vessel [41], but the blood within membrane feeders is often static while the different intensity of pressure or the hardness of the membrane itself may increase the difficulty for mosquitoes to penetrate. Therefore, there is a risk that *Ae. aegypti* maintained on membrane feeders will lose their attraction to host cues and have a reduced ability to feed on live human hosts. Adaptation to membrane feeding may affect experimental outcomes, particularly for studies involving host-seeking, feeding behavior and repellency. Adaptation may also reduce the quality of mosquitoes reared for field release since

host-seeking and feeding ability are critical to the success of population replacement programs.

Here, we test adaptation to membrane feeding by selecting replicate populations of *Ae. aegypti* for feeding on two blood sources: human arms or membrane feeders. We then evaluate their host-seeking ability, attraction to host cues, feeding ability and life history traits.

## Methods

### Ethics statement

Blood feeding of female mosquitoes on human volunteers for this research was approved by the University of Melbourne Human Ethics Committee (approval 0723847). All adult subjects provided informed written consent (no children were involved).

### Mosquito strains and colony establishment

Two populations of *Aedes aegypti* mosquitoes established in the laboratory five years apart were chosen for experiments. A recently-derived "field" population was established from eggs collected by ovitraps that were placed around Yorkeys Knob, Far North Queensland, Australia in February 2018. No permits were required for mosquito collections; verbal permission was obtained from each household before setting up traps. *Aedes aegypti* were identified as larvae using a key [42] and other species were discarded. Approximately 450 larvae were reared to adulthood, females were blood-fed on a human volunteer, and the resulting progeny were used to establish replicate populations for experiments. An older "laboratory" population had been established from *Aedes aegypti* originating from Gordonvale and Yorkey's Knob, Far North Queensland, Australia in May 2013. This population had been maintained in the laboratory for approximately 60 generations when the field population was established. Both populations were infected with the *w*Mel strain of *Wolbachia* as they were collected from locations where *w*Mel-infected *Ae. aegypti* were released in 2011 [43]. Before the adaptation experiments commenced, mosquitoes were maintained according to Ross et al. [16] in 19.7 L (27 cm$^3$) Bug-Dorm-1® cages (MegaView Science Co., Ltd., Taichung City, Taiwan), with females being blood-fed on a human volunteer each generation.

### Selection for feeding on human arms and membrane feeders

Progeny from the "laboratory" and "field" populations were each divided into four populations. Two field and two laboratory populations were fed on human arms each generation, while the remaining four were fed on membrane feeders. A schematic showing the establishment and selection of the eight populations on the two blood sources is shown in Fig 1A. The eight populations were maintained separately in BugDorm-1® cages at a census size of 400 individuals each generation. Populations underwent selection for at least 11 generations on the two blood sources before commencing experiments, except for the host-seeking experiment in the semi-field cage which was conducted after 8 generations of selection. Life history experiments (including fecundity, egg hatch, larval development time, survival to adulthood and wing length) were conducted after 11 generations of selection, host-seeking behavior (in a laboratory cage), feeding proportion, blood meal weight and feeding duration were tested after 12 generations of selection, while experiments testing attraction to host cues were conducted after 13 generations of selection.

Populations selected for feeding on human arms were given blood meals according to Ross et al. [16]. Females (5–7 d old and sugar-starved for 24 hr) were provided access to a bare human forearm for 15 min or until all females had fed to repletion (Fig 1B). The same human

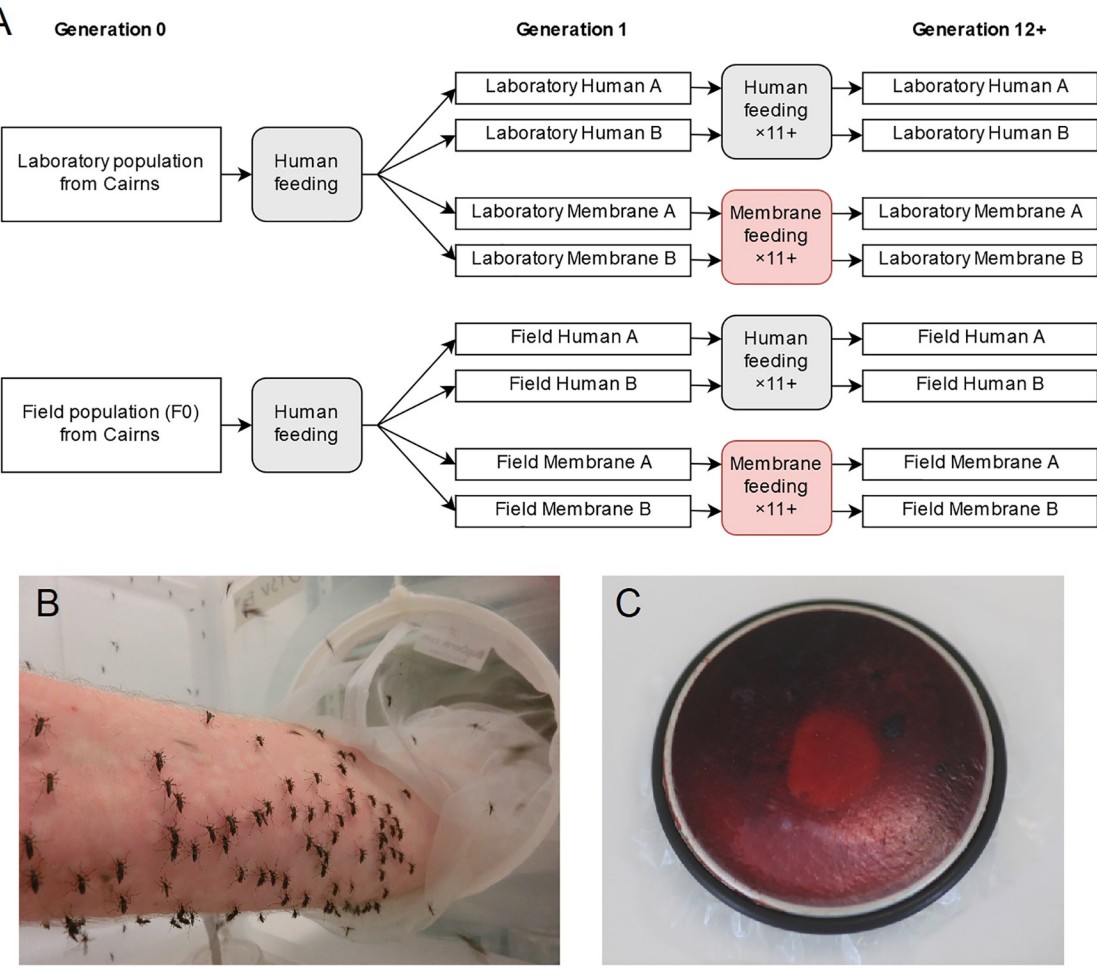

**Fig 1. Establishment and selection of the laboratory and field *Aedes aegypti* populations for feeding on human arms or membrane feeders.** (A) The eight populations underwent selection for at least 11 generations on each blood source before conducting experiments (except for the semi-field experiment). (B) Direct human arm blood feeding. (C) Completed membrane feeding apparatus, showing a blood-filled reservoir covered with a collagen membrane and sealed by a rubber ring (image credit: Veronique Paris).

volunteer was used for all experiments. For populations selected for feeding on membrane feeders, females (5–7 d old and sugar-starved for 24 hr) were provided access to human blood *via* a membrane feeder for 1 hr. Human blood was sourced from the Red Cross (Agreement #16-10VIC-02) once per month and kept at 4°C. The same batch of blood from a single anonymous donor was used for all populations within generations and experiments, but batches differed between experiments and across generations. A 6 mL Hemotek reservoir (Discovery Workshops, Accrington, UK) covered with a sheet of collagen feeding membrane (Discovery Workshops, Accrington, UK) and sealed by a rubber ring was filled with blood using a syringe and plugged with nylon stoppers (Fig 1C). A pocket hand warmer (Kathmandu) was placed over the feeder to heat it. The heated feeder and hand warmer were then placed on top of the cage, membrane side down, and left for one hour to allow females to feed.

## Life history parameters

We compared life history parameters of laboratory- and field-derived mosquitoes after being selected for feeding on the two blood sources for 11 generations. Eggs from each population

were hatched in plastic trays filled with 3 L of reverse osmosis (RO) water and provided with a few grains of yeast and a crushed TetraMin® tropical fish food tablet (Tetra, Melle, Germany). Within five hours of hatching, larvae were counted into 700 mL plastic trays filled with 500 mL of RO water and provided with TetraMin *ad libitum* until pupation. Six trays of 100 larvae were reared for each of the eight populations. Pupae were counted twice per day (in the morning and evening) to measure development time and adults from the six replicate trays were released into a single BugDorm-1® cage. The number of adults emerging from each tray was recorded to calculate the proportion of larvae that survived to adulthood.

After all adults had emerged, twenty females and twenty males were selected at random from each cage. Wing length of these individuals was measured according to Ross et al. [44] using the distance from the alular notch to the wing tip (excluding fringe scales). Damaged or folded wings (~5% of wings) were excluded from the analysis.

One week after adults had started to emerge, 80 females were selected at random from each population and aspirated into two 3 L cages with 40 females each. Females were held without food (10% sucrose solution) for 24 hr. One cage from each population was then provided with the forearm of a human volunteer for 10 min, while the other cage was provided with a membrane feeder with human blood for 10 min, as described above. Twenty females that fed on a human volunteer and 20 females that fed on the membrane feeder from each population and were visibly engorged were isolated for oviposition. If fewer than 20 females fed on the membrane feeder, a second cage of 40 females was set up with a new membrane feeder and left for 30 min. Females were isolated in 70 mL specimen cups covered with a mesh lid, filled with 20 mL of larval rearing water and lined with a sandpaper strip as an oviposition substrate. Four days after blood feeding, eggs were collected, partially dried, maintained at a high humidity and then hatched three days after collection. Cups were checked every second day for an additional week, and any eggs were collected and hatched three days after collection. Egg hatch proportions were determined by dividing the total number hatched eggs (where the egg cap is clearly detached) by the total number of eggs laid for each female. Females that did not lay eggs (~15% of females) were excluded from the analysis.

## Blood meal weight and feeding duration

After 12 generations of selection, we compared populations for their feeding duration and blood meal weight when fed on a human host. A 30 mL pill cup with a mesh lid was weighed on an analytical balance (Sartorius BP 210 D). A female (7 d old, mated and sugar-starved for approximately 24 hr) was aspirated into the cup and the cup was weighed again to determine the fresh weight of the female. The mesh lid was then pressed against the right palm of a human volunteer to provide the female access to blood. Blood feeding duration was timed with a Jastek digital timer from when the stylet pierced the skin to when the proboscis was withdrawn. Therefore, probing time (the time after the proboscis was inserted but before ingestion) was included in the duration. If the female did not attempt to feed within 5 min, the female was discarded and replaced. The cup and the cup plus mosquito were weighed again, and blood meal size was calculated by subtracting the weight of the unfed female from the weight of the engorged female. Experiments were conducted across three days with separate groups of 7 d old females, with the same human volunteer used for all experiments. Five females from each population were tested each day for their blood feeding duration and blood meal weight. However, due to an error with the balance in one experiment, we only obtained 10 measurements of blood meal weight rather than 15 measurements for each population. We did not test blood meal weight and feeding duration with females fed on membrane feeders; as membrane feeders are heated, both blood consistency and membrane texture changed over

time and these changes were expected to influence feeding duration, making population comparisons difficult.

## Feeding proportion

We examined the proportion of female mosquitoes after 12 generations of selection that blood-fed successfully on a membrane feeder or human volunteer. Forty females (7 d old, mated, and sugar-starved for approximately 24 hr) from each population were aspirated into 3 L cages. One cage per population was provided with a human forearm and one cage was provided with a membrane feeder (see "Selection for feeding on human arms and membrane feeders") for 10 minutes. The proportion of females that fed was determined by dividing the number of visibly engorged females by the total number of mosquitoes in the cage. This experiment was repeated on three separate days, with one cage provided with a human forearm and one cage provided with a membrane feeder each day for the eight populations. A single human volunteer and a single source of blood was used for all human forearm and membrane feeder treatments respectively.

Since feeding proportions were variable in the membrane feeder treatments due to the reasons mentioned above, we conducted a second experiment where two populations (human- and membrane-selected) were tested in the same cage to control for differences between membrane feeders. Twenty human-selected and 20 membrane-selected females (7 d old, mated, and sugar-starved for approximately 24 hr) were aspirated into a 3 L cage and then provided with a membrane feeder for 10 min. The human-selected population was paired with the membrane-selected population of the same replicate (A or B) and origin (field or laboratory). For instance, the Field Human B population was paired with the Field Membrane B population. Since the human- and membrane-selected populations could not be distinguished phenotypically, the populations were lightly marked with fluorescent powder (DayGlo powder, Barnes Products, Moorebank, Australia) as described in Lau et al. [45]. This involved aspirating females into a 70 mL cup containing 0.3 mg of green or orange powder which was then gently shaken to mark the mosquitoes. Populations were identified under UV light, and colors were swapped between experiments. Feeding proportions were determined for six replicate cages for each paired set of populations.

## Host-seeking

We compared the ability of human-selected and membrane-selected females to seek human hosts in small laboratory cages and in a semi-field enclosure. Experiments in laboratory cages were performed after 12 generations of selection. In the laboratory experiments, five females (7 d old, mated, and sugar-starved for approximately 24 hr) were aspirated into a BugDorm-1® cage and allowed to acclimate for five minutes. A volunteer then inserted a bare arm though a mesh stocking in the front of the cage and placed their hand on the bottom of the cage in the center. A second person recorded the duration to landing for each female until all mosquitoes had landed or 5 min had elapsed. Durations were timed from when the hand was placed on the bottom of the cage to when females landed on bare skin and remained there for at least two seconds. Two experiments were conducted simultaneously with the left and right arms of a single volunteer, one with a membrane-selected population and the other with a human-selected population, and sides were swapped between replicates. Experiments were repeated ten times for "A" replicate populations and five times for "B" replicate populations.

The semi-field experiments followed methods described in Lau et al. [45] and were conducted in an enclosure designed to simulate a typical yard and Queenslander-style house understory [46]. Experiments in the semi-field cage were performed after 8 generations of

selection. Fifty membrane-selected (Field membrane A) and 50 human-selected (Field human A) females (7 d old, mated, and sugar-starved for approximately 24 hr) were dusted with fluorescent powder (see "Feeding proportion"). The two groups were mixed by releasing them into a 5 L plastic container with a mesh cover and the container was placed at one end of the semi-field enclosure where the mosquitoes were left to acclimate for at least 10 min. Two experimenters (one female, one male) sat on the ground within the understory at the other end of the enclosure, 15 m away from the container and 1 m apart from each other. The experimenters exposed their lower legs but wore gloves, tops with long sleeves and hats with a mesh veil to deter mosquitoes from landing elsewhere. At the beginning of each experiment, mosquitoes were released from the plastic container and mosquitoes that landed on exposed skin were collected using mechanical aspirators (BioQuip Products Inc. flashlight aspirator). The collection cup was replaced with an empty cup every three minutes until 42 min had elapsed. The number of mosquitoes from each population in each cup was counted to estimate the median time to arrival. The proportion of mosquitoes that landed during the experiment was estimated by dividing the total number of mosquitoes collected by the number of mosquitoes released. The experiment was repeated three times, and a combination of mechanical aspirators, BG-Sentinel traps (BioGents, Regensburg, Germany) and electric rackets were used to deplete the semi-field cage of mosquitoes between experiments.

### Attraction to host cues

We compared the response of human-selected and membrane-selected populations to host cues in a two-port olfactometer, conducted after 13 generations of selection. We constructed the olfactometer from a BugDorm-1® cage with a similar design to a previous study [47]. One wall of the cage was removed and replaced with a thick plastic sheet that was connected to two funnels leading to cylindrical traps (9 cm long, 8 cm inner diameter, BioQuip Products Inc. mini mosquito breeder). The centers of the traps were 16 cm apart. A stimulus port identical in size to the trap was attached to each trap. The traps and stimulus ports both had holes in one end (4.5 cm diameter) that were covered in mesh to prevent mosquitoes from escaping but allowing air to flow. A box fan placed at the opposite end of the cage drew air through the two ports into the cage. The rate of air flow in each port was ~0.2 m/s as measured by a Kestrel 2000 air velocity meter (Kestrel, East Melbourne, Australia).

Before each experiment, 20 females (7 d old, mated, and sugar-starved for approximately 24 hr) from a human-selected population and 20 females from a membrane-selected population were dusted with fluorescent powder (see "Feeding proportion") and then mixed in a 500 mL plastic container. As per the feeding proportion experiments, the human-selected population was paired with the membrane-selected population of the same replicate (A or B) and origin (field or laboratory).

We tested attraction to three stimuli: heat, human odor and a human hand. To test attraction to heat, a pocket hand warmer (Kathmandu) was activated and placed in one stimulus port, while the other port was left empty. To test attraction to human odors, a sock, worn for approximately 4 hr by a single human volunteer, was placed in the stimulus port, while the other port contained an identical, but unworn sock. To test attraction to a human hand, the palm of a single human volunteer was held 1 cm in front of the mesh at the end of one stimulus port, while the other port was blank.

At the start of the experiment the stimulus was placed in the stimulus port (or a hand was held next to the port for the human hand treatments) and the fan was turned on. Mosquitoes were then released into the cage. After 10 minutes, the funnels to both traps were closed and the number of mosquitoes from each population in each trap was recorded. For each

population, the proportion of mosquitoes attracted to the stimulus was determined by dividing the number of mosquitoes in the stimulus trap by the total number of mosquitoes tested. Mosquitoes that were visibly damaged were excluded from the analysis. Between each experiment the stimulus port and control port were swapped. Experiments testing attraction to a human hand, heat and human odor were repeated 14, 16 and 17 times respectively. For each stimulus, the four pairs of populations were tested 3–5 times each.

## Statistical analysis

All data were analyzed using SPSS Statistics version 24.0 for Windows (SPSS Inc, Chicago, IL). Data sets were tested for normality with Shapiro-Wilk tests and transformed where appropriate. Data for time to landing in BugDorm cages and feeding duration failed Shapiro-Wilk tests but were normally distributed following log transformation, while angular transformation was used for data for feeding proportion, survival to adulthood and egg hatch proportion. Data that were normally distributed were analysed with general linear models (GLMs). We tested for effects of sex, population origin (laboratory or field), blood source on which populations had been selected (human or membrane) and replicate population (A or B). Replicate population (nested within blood source × population origin) and experiment date were included as random factors. In some cases, proportions were not normally distributed even after angular transformation, but in these situations we still used GLMs to test for the importance of factors after averaging proportions for each replicate population (which provided the denominator for F tests in the GLMs). Feeding proportions in mixed cohorts and attraction to control ports in a two-port olfactometer were analysed with Wilcoxon signed-rank tests (where observations from the lines came from the same cage based on marked mosquitoes). We used log-rank tests to compare cumulative landing proportions between human-selected and membrane-selected populations in laboratory and semi-field cage host-seeking experiments.

# Results

## Life history parameters

We compared life history parameters of laboratory- and field-derived populations after being selected for feeding on human arms or membrane feeders. Females were substantially slower to develop than males (GLM: $F_{1,85} = 196.999$, $P < 0.001$) and had larger wings ($F_{1,280} = 5470.768$, $P < 0.001$, Table 1); we therefore analysed the sexes separately.

**Table 1. Larval development time, survival to adulthood and wing length of *Aedes aegypti* populations derived from the laboratory and field and selected for feeding on human arms or membrane feeders.**

| Population | Larval development time (days)* | | Survival to adulthood (proportion)* | Wing length (mm)* | |
|---|---|---|---|---|---|
| | Females | Males | | Females | Males |
| Laboratory Human A | 6.18 (6.06, 6.27)[a] | 5.85 (5.70, 5.99)[a] | 1.00 (0.99, 1.00)[a] | 2.96 (2.94, 2.99)[a] | 2.25 (2.21, 2.28)[ab] |
| Laboratory Human B | 6.17 (6.11, 6.36)[ab] | 5.84 (5.77, 6.01)[a] | 0.99 (0.98, 1.00)[a] | 2.96 (2.91, 3.00)[a] | 2.23 (2.20, 2.25)[abc] |
| Field Human A | 6.31 (6.28, 6.39)[ab] | 6.00 (5.93, 6.09)[ab] | 0.99 (0.98, 1.00)[a] | 2.99 (2.95, 3.02)[a] | 2.27 (2.25, 2.29)[a] |
| Field Human B | 6.28 (6.14, 6.44)[ab] | 5.88 (5.83, 5.90)[a] | 0.98 (0.97, 0.99)[a] | 2.91 (2.90, 2.99)[a] | 2.21 (2.19, 2.24)[abc] |
| Laboratory Membrane A | 6.39 (6.29, 6.66)[bc] | 6.18 (6.08, 6.33)[c] | 0.99 (0.97, 1.00)[a] | 2.97 (2.94, 3.05)[a] | 2.21 (2.19, 2.23)[abc] |
| Laboratory Membrane B | 6.66 (6.40, 6.81)[c] | 6.27 (6.16, 6.32)[c] | 0.97 (0.92, 1.00)[a] | 2.86 (2.83, 2.96)[a] | 2.17 (2.14, 2.19)[c] |
| Field Membrane A | 6.29 (6.19, 6.45)[ab] | 6.09 (5.99, 6.16)[bc] | 0.97 (0.93, 0.99)[a] | 2.91 (2.85, 2.96)[a] | 2.23 (2.18, 2.28)[abc] |
| Field Membrane B | 6.38 (6.27, 6.52)[abc] | 6.01 (5.93, 6.08)[ab] | 0.99 (0.96, 1.00)[a] | 2.94 (2.89, 3.01)[a] | 2.19 (2.15, 2.22)[bc] |

*Medians are shown followed by lower and upper 95% confidence intervals in parentheses. Within a column, different letters indicate significant (P < 0.05) differences between populations by Tukey's post-hoc test with Bonferroni correction for the number of traits compared.

Populations selected for membrane feeding were slower to develop (by 3.5% on average) than human-selected populations (GLM: females: $F_{1,4} = 19.064$, P = 0.012, males: $F_{1,4} = 30.510$, P = 0.005). There was no effect of population origin or replicate population on development time for either sex (all P > 0.05). However, there was a significant interaction between blood source and population origin for both females ($F_{1,4} = 10.985$, P = 0.029) and males ($F_{1,4} = 8.955$, P = 0.040), with larger effects of blood source on development time in the laboratory populations.

Survival to adulthood did not differ between human- and membrane-selected populations, with no effect of population origin or replicate population (GLM: all P > 0.05). Wing length was unaffected by blood source or population origin in both sexes, with no interaction between blood source and population origin (all P > 0.05). However, there was an effect of replicate population in both sexes (females: $F_{4,136} = 3.387$, P = 0.011, males: $F_{4,134} = 4.298$, P = 0.003). Overall, membrane feeding across generations influenced development negatively but had no impact on survival or wing size.

We tested the fertility of human- and membrane-selected populations when females were fed on human arms or membrane feeders (Fig 2). When fed on human arms, females selected for membrane feeding exhibited lower fecundity (by 13.4% on average) than human-selected populations (GLM: $F_{1,4} = 37.410$, P = 0.003, Fig 2A), with no effect of population origin or replicate population (all P > 0.05). There was a significant interaction between population origin and blood source ($F_{1,4} = 27.271$, P = 0.006), with larger effects of blood source on fecundity in the field populations. In contrast, there was no effect of blood source, population origin or replicate population and no interaction between population origin and blood source when females were fed on membrane feeders (all P > 0.05, Fig 2B). Females tended to have lower fecundity when fed on membranes ($F_{1,265} = 25.897$, P < 0.001) which may reflect a reduction in quality due to blood storage. However, a direct comparison between the two blood sources cannot be made since the blood was derived from two different humans and mosquito fecundity is known to differ between human volunteers [19].

In contrast to fecundity, egg hatch proportions did not differ significantly between human- and membrane-selected populations when fed on human arms (GLM: $F_{1,4} = 5.445$, P = 0.080, Fig 2C) or membrane feeders ($F_{1,4} = 0.013$, P = 0.913, Fig 2D), with no effects of population origin (all P > 0.05). However, there was a significant effect of replicate population when mosquitoes were fed on human arms ($F_{1,124} = 2.833$, P = 0.027). Egg hatch proportions also did not differ between the two blood sources ($F_{1,265} = 0.010$, P = 0.919). The subtle but consistent fitness costs in membrane-selected populations likely reflect inbreeding depression [14] arising from bottlenecks each generation due to low feeding proportions (see "Feeding proportion" results).

## Blood meal weight and feeding duration

We tested if populations selected for feeding on membrane feeders differed from human-selected populations in their ability to ingest blood from a human volunteer. Log feeding duration did not differ between populations, with no effect of population origin, blood source or replicate population according to a GLM (all P > 0.05, Fig 3A). However, feeding duration was affected by experiment date ($F_{2,110} = 6.606$, P = 0.002), with the third day showing shorter feeding durations despite females being the same age in each experiment.

Blood meal weight was unaffected by the population origin, blood source or the date of the experiment (GLM: all P > 0.05), but there was a significant effect of replicate population ($F_{4,71} = 2.913$, P = 0.027, Fig 3B), where populations A and B of some treatments differed in blood meal weight.

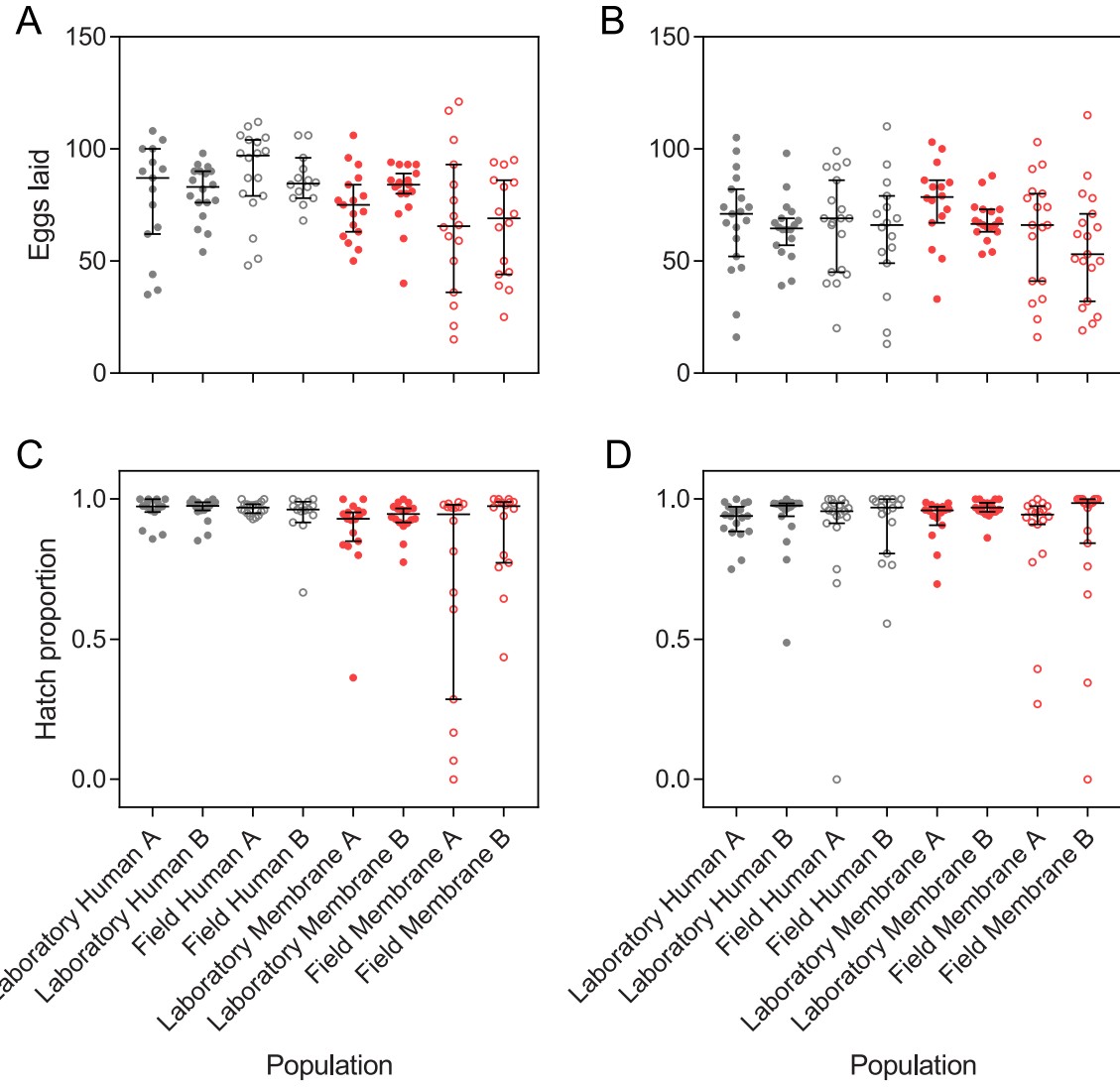

**Fig 2. Fecundity (A, B) and egg hatch proportions (C, D) of *Aedes aegypti* populations derived from the laboratory and field populations and selected for feeding on human arms or membrane feeders.** All populations were then fed on either human arms (A, C) or human blood through a membrane feeder (B, D) and isolated for oviposition. Bars are medians with 95% confidence intervals.

## Feeding proportion

To see if membrane-selected populations had maintained their attraction to and ability to feed on humans, we tested the proportion of females that fed on human arms. Human- and membrane-selected females both exhibited high feeding proportions on human arms (Fig 4A), with no effect of blood source, population origin or replicate population (GLM: all P > 0.05) but a significant effect of experiment date ($F_{2,14} = 5.079$, P = 0.022). This suggests that membrane-selected populations have maintained their feeding ability and attraction to humans.

After 12 generations of selection, we expected the membrane-selected populations to adapt to feeding on membranes through an improvement in the proportion of females feeding to repletion on membrane feeders. Feeding proportions on membranes were lower and more variable than on humans (Fig 4B). Membrane-selected populations exhibited higher feeding proportions on membranes than human-selected populations (GLM: $F_{1,4} = 17.753$, P = 0.014),

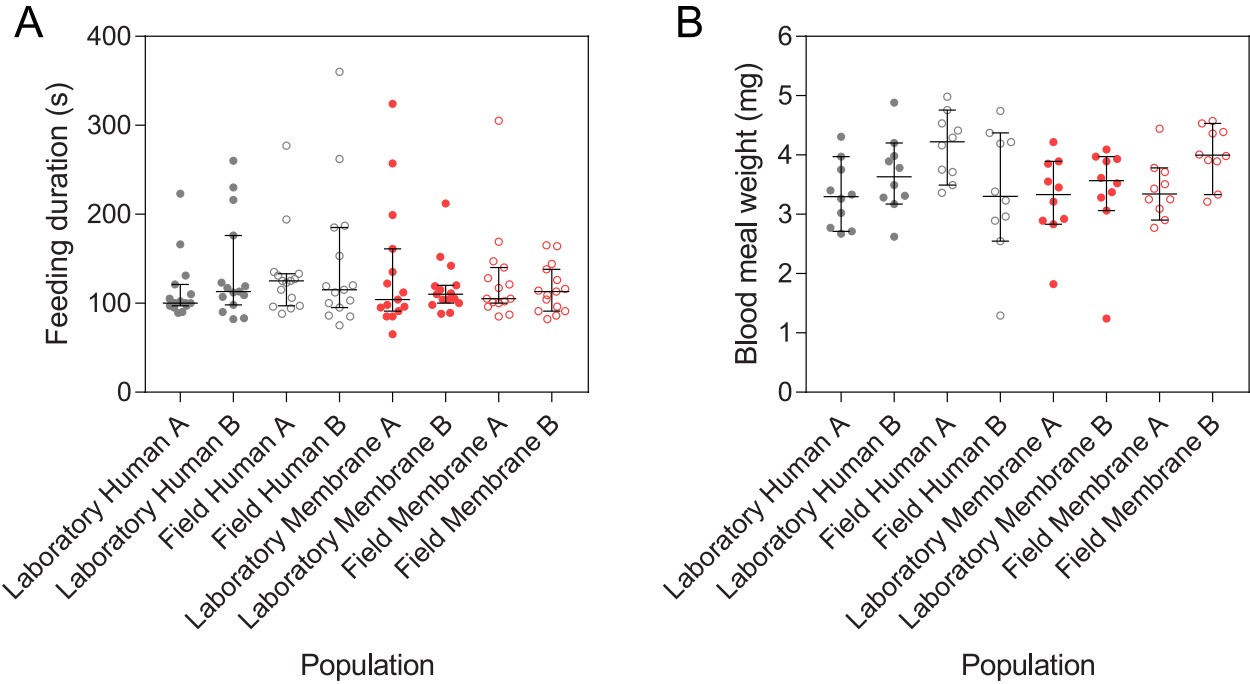

**Fig 3. Feeding duration (A) and blood meal weight (B) of membrane- and human-selected *Aedes aegypti* populations when females were fed on a human volunteer.** Bars are medians with 95% confidence intervals.

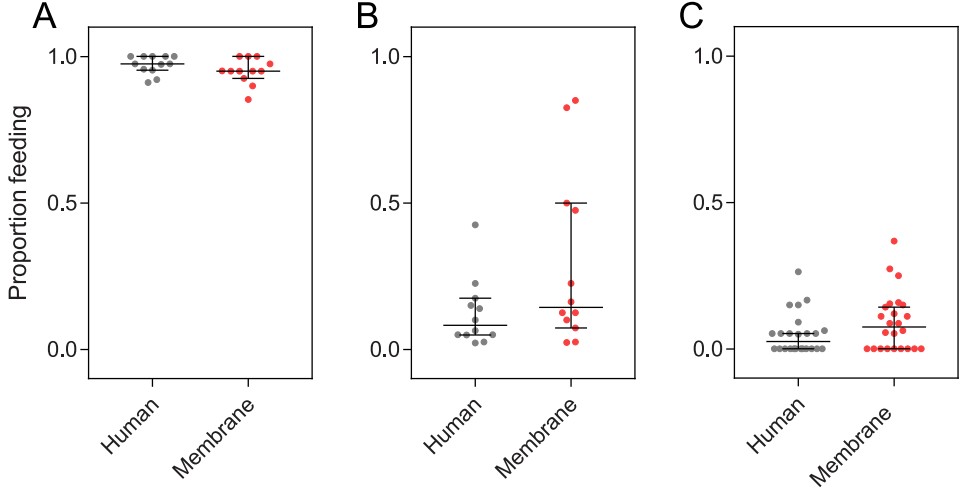

Blood source for selection

**Fig 4. Proportion of *Aedes aegypti* females selected on human arms or membrane feeders that were visibly engorged when provided access to a human arm (A) or membrane feeder (B,C) for 10 min.** For experiments with membrane feeders, human-selected and membrane-selected populations were tested in separate cages (B) or in mixed cohorts in the same cage (C) where populations were marked with different colors of fluorescent powder. Data for all four populations selected for feeding on each blood source were pooled. Bars are medians and 95% confidence intervals.

an increase of 138% on average, indicating adaptation to membrane feeding. There was no effect of population origin or replicate population and no interaction between population origin and blood source (all P > 0.05) but there was a significant effect of experiment date ($F_{2,14}$ = 10.180, P = 0.002).

To control for potential differences between membrane feeders, we also tested feeding proportions when human- and membrane-selected females were mixed in the same cage. In this experiment there was also a significantly higher proportion of membrane-selected females feeding than human-selected females (Wilcoxon signed-rank test: Z = 1.988, P = 0.0466), an increase of 82.4% on average.

### Host-seeking

We tested the host-seeking ability of human- and membrane-selected females by measuring the time to landing on human hosts in laboratory and semi-field cages. The cumulative proportions of human- and membrane-selected females landing over time in laboratory cages were not significantly different according to a Log-rank test ($\chi^2$ = 1.402, df = 1, P = 0.236, Fig 5A). Females from both sets of populations were quick to land on the host in the laboratory

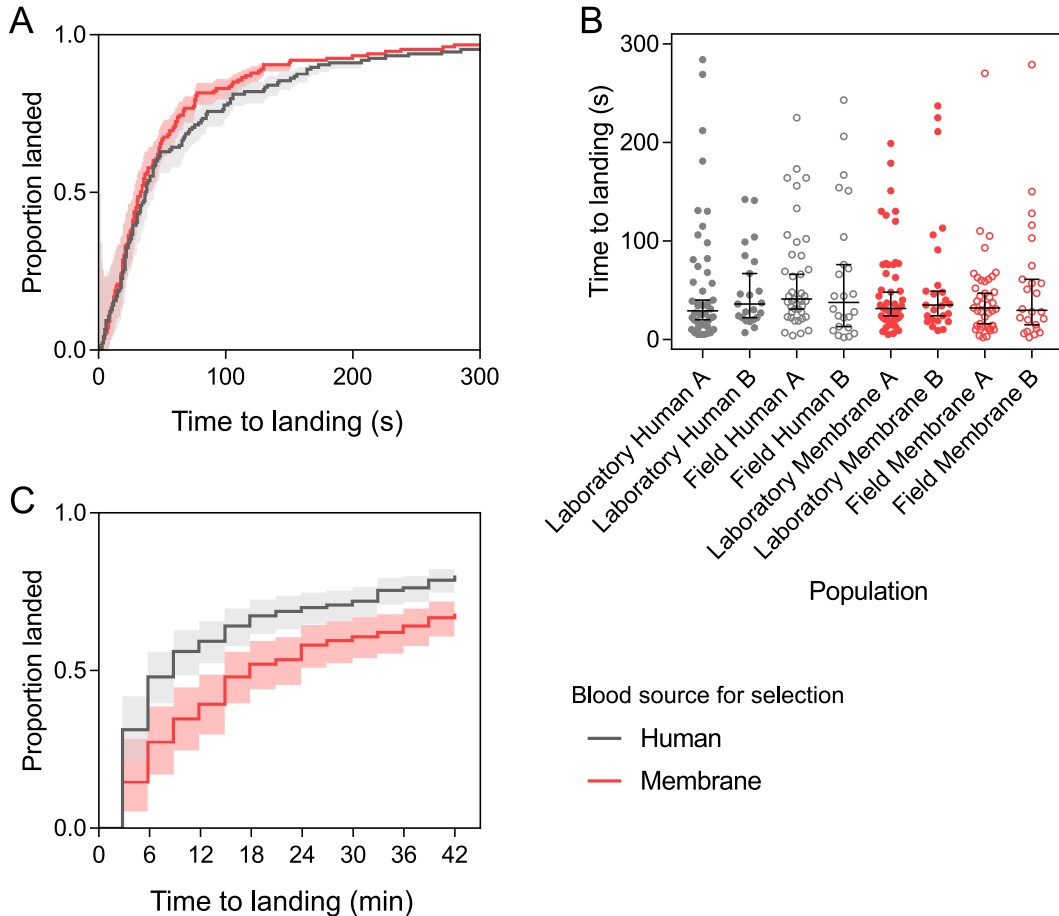

**Fig 5. Host-seeking ability of human-selected and membrane-selected *Aedes aegypti* females in BugDorm cages in the laboratory (A-B) and in a semi-field cage (right).** (A) and (C) show the cumulative proportion of human- and membrane-selected females landing over time, where shaded areas are 95% confidence intervals. Landing times for individual mosquitoes from each population tested in laboratory cages are shown in (B) with medians shown, where error bars are 95% confidence intervals.

cage, with almost all females landing within 5 min. Population origin, blood source and replicate population had no significant effect on the (log) time to landing in laboratory cages, with no interaction between population origin and blood source (GLM: all P > 0.05, Fig 5B).

Females were much slower to land in semi-field cages where we tested host-seeking over a longer distance. In semi-field cages, it took 18 minutes for 50% of females from the membrane-selected population (Field Membrane A) to land, while the human-selected population (Field Human A) took 9 minutes (Fig 5C), though this difference was not significant (GLM: $F_{1,2} = 4.568$, P = 0.166). Cumulative landing proportions over time differed between human- and membrane-selected populations according to a Log-rank test ($\chi^2 = 10.01$, df = 1, P = 0.002), where human-selected populations were quicker to land. However, there was also an effect of experimental replicate ($\chi^2 = 14.263$, df = 2, P = 0.001) since landing times differed substantially between experiments. The median time to landing was also affected by strain (GLM: $F_{1,2} = 49.00$, P = 0.020) and experimental replicate ($F_{2,2} = 73.00$, P = 0.014) but the total proportion of females landing after 42 min did not differ significantly between human- (mean ± SD = 0.80 ± 0.07) and membrane-selected (0.68 ± 0.14) females ($F_{1,2} = 1.895$, P = 0.303).

### Attraction to host cues

Since membrane-selected females maintained their host-seeking ability when tested in laboratory cages but seemed impaired when tested in semi-field cages, we tested if females selected on the two blood sources differed in their attraction to separate host cues. Because membrane feeders provide visual and thermal but not olfactory stimuli, we expected membrane-selected mosquitoes to maintain their attraction to heat packs (as a proxy for heat) but potentially lose their attraction to worn socks (as a proxy for human odor). We therefore tested this in a two-port olfactometer (Fig 6A).

There was no effect of population origin or replicate population, with no interaction between blood source and population origin in all three experiments (GLM: all P > 0.05). Human- and membrane-selected populations did not differ in their attraction to worn socks (GLM: $F_{1,4} = 1.887$, P = 0.241; Fig 6C), However, more membrane-selected females were attracted to human hands than human-selected females ($F_{1,4} = 14.235$, P = 0.020), and there was also a marginally significant difference for attraction to heat packs ($F_{1,4} = 7.127$, P = 0.052). Low proportions of females were collected from the control ports in all experiments and there was no effect of selection (Wilcoxon signed-rank test: all P > 0.05, Fig 6B).

### Discussion

We tested whether *Ae. aegypti* adapt to feeding on blood through artificial membranes across generations and if this affected mosquito performance. While membrane-selected populations suffered costs to life history traits, they maintained their attraction to host cues and feeding ability on humans. Recent field release programs have used blood provided *via* membranes for mass-rearing *Ae. aegypti* [4, 48] and our results indicate that using membrane feeders rather than live humans will probably not substantially compromise mosquito performance. Furthermore, maintaining *Ae. aegypti* in the laboratory on membrane feeders may not influence the outcomes of experiments where lab-reared mosquitoes are taken to be representative of wild mosquitoes. However, the source of blood and the type of membrane feeder should be carefully considered; some blood sources, particularly non-human blood for maintaining *Ae. aegypti*, can reduce mosquito performance [18, 21].

In our experiments, membrane-selected populations had extended development and reduced fertility relative to human-selected populations. These costs likely represent

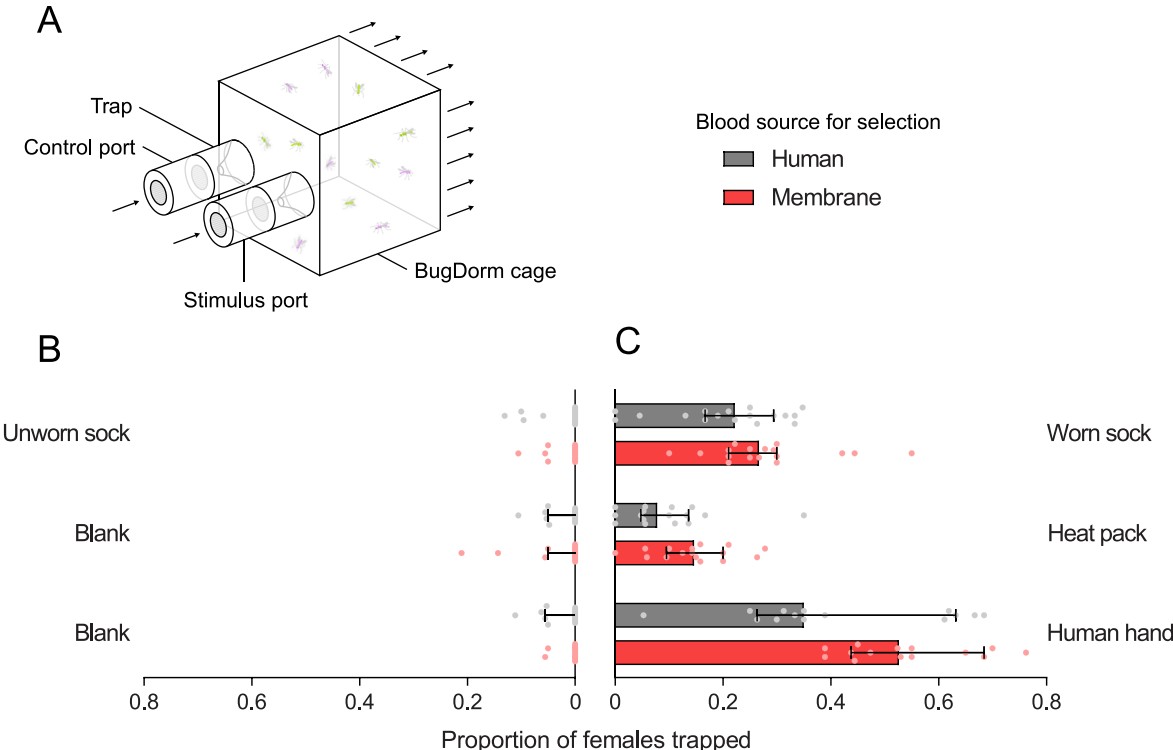

**Fig 6. Attraction of female *Aedes aegypti* selected on human arms or membrane feeders to host cues in a two-port olfactometer.** (A) Schematic of olfactometer showing the location of the stimulus and control ports (which were alternated between experiments). Arrows indicate the direction of airflow through the olfactometer. (B-C) The proportion of mosquitoes trapped in the control port (B) and stimulus port (C) for human-selected (gray) and membrane-selected (red) populations. Bars show median proportions and 95% confidence intervals while dots show proportions for individual trials.

inbreeding depression resulting from membrane-selected populations passing through bottlenecks each generation. Though membrane-selected and human-selected populations were maintained at the same census size, feeding rates on membranes were poor relative to human arms where typically >95% of females successfully took a blood meal. Therefore, the effective population size of the membrane-selected populations is likely to be much lower than in the human-selected populations [14] and population sizes for stock maintenance will need to be increased if inbreeding depression is to be avoided.

We conducted experiments with populations derived from the laboratory and field to test for effects of laboratory maintenance. Despite the laboratory and field populations being colonized 5 years apart, we found no effect of population origin in any of our experiments. This suggests that long-term laboratory maintenance has little influence on life history, attraction to humans or feeding ability, consistent with a study comparing the fitness of a near-field population to populations maintained in the laboratory for one year [14]. However, since we reared both populations in the laboratory for several generations before experiments commenced, we were not able to test for rapid adaptation. In *Ae. aegypti*, traits such as blood feeding duration can change markedly within only a few generations of laboratory maintenance [49]. We did find differences between replicate populations for wing length, egg hatch proportion and blood meal weight which may reflect drift [14] or rapid adaptation in some lines. The differences that developed between replicate populations emphasize that any evaluation of life history effects due to line modification (such as by *Wolbachia* or genetic modification) should

ideally be carried out with independent replication and/or careful control of the genetic background (e.g. through introgression).

We found evidence of adaptation to membrane feeding, with membrane-selected females having greater feeding success than human-selected populations when fed on membranes. However, overall feeding rates on membranes were low and variable in both sets of populations, which may be explained by an inability of mosquitoes to reach the blood rather than a lack of attraction to the feeding apparatus. During population maintenance we observed that while most females landed on the surface of the feeder, many could not piece the membrane with their proboscis (S1 Video). Although we did not compare different types of membranes in this study, some materials are more suitable for blood feeding than others [34, 35]. Where there are issues with low feeding rates, as occurred here, researchers should be careful to avoid bottlenecks and keep populations large during maintenance. During mass-rearing, egg production may be improved by choosing a membrane that is easily pierced, providing more time or a larger surface area for blood feeding, or by rubbing the membrane on skin to increase attraction to the feeder [29]. Supplementing blood with ATP may also increase feeding rates as demonstrated in *Aedes albopictus* [50].

*Aedes aegypti* locate hosts through the detection of thermal, visual and olfactory cues [37, 38, 40]. With the alteration of selective pressures during membrane feeding, we hypothesized that membrane-selected populations may lose their attraction to human odors but maintain their attraction to heat. Membrane- and human-selected populations showed similar attraction to human odor and heat and had similar landing rates on a human arm in laboratory experiments, demonstrating that host-seeking is not compromised over short distances. In contrast, membrane-selected females were slower host-seekers in the semi-field cage. Host-seeking over longer distances relies on the detection of $CO_2$ [51] and visual contrast [37, 52] and will depend on flight ability, which we did not test directly in our experiments. While the differences between populations may reflect adaptation, the result could be confounded by inbreeding or drift given that we only tested a single human-selected and membrane-selected population in this experiment [14]. With the properties of membrane feeders being different to human skin, we also hypothesized that membrane-selected populations would exhibit a reduced ability to feed on humans, but found no effect of selection on feeding duration, blood meal weight or proportion feeding on humans. However, as demonstrated by our host-seeking experiments, costs to blood feeding ability may be apparent under more realistic conditions, such as when mosquitoes are provided with multiple blood meals. Although shifts in these traits may occur over longer periods, field releases do not often involve old laboratory stocks.

Together, these results suggest that membrane feeding by itself will not directly compromise the quality of *Ae. aegypti* mosquitoes, even after one year of laboratory maintenance. While there is some evidence of adaptation to membrane feeding, the effects were quite small and this did not affect feeding ability or attraction to humans. Hence mass-rearing procedures that rely on membrane feeding devices for egg production can be expected to produce females that nevertheless remain effective at feeding on human hosts after release. Although we observed deleterious effects of membrane feeding that may affect the outcomes of field releases, outcrossing release stocks to field populations will likely alleviate these effects [14, 53].

## Supporting information

**S1 Video. Examples of *Aedes aegypti* blood feeding behavior on a Hemotek feeder and collagen membrane illustrating low feeding success.**
(MP4)

**S1 Appendix. General linear models for all parameters tested in the study.**
(DOCX)

## Acknowledgments

We thank Chengjun Li for technical assistance, Jason Axford for sourcing the blood for membrane feeding and Ellen Cottingham, Veronique Paris and Ashley Callahan for establishing the membrane feeding methods. We also thank Scott Ritchie's group at James Cook University Cairns for providing mosquito eggs for colony establishment. Finally, we thank two anonymous reviewers for their constructive feedback on an earlier version of the manuscript.

## Author Contributions

**Conceptualization:** Perran A. Ross, Ary A. Hoffmann.

**Formal analysis:** Perran A. Ross.

**Funding acquisition:** Ary A. Hoffmann.

**Investigation:** Perran A. Ross, Meng-Jia Lau.

**Methodology:** Perran A. Ross, Meng-Jia Lau, Ary A. Hoffmann.

**Supervision:** Ary A. Hoffmann.

**Visualization:** Perran A. Ross.

**Writing – original draft:** Perran A. Ross.

**Writing – review & editing:** Perran A. Ross, Meng-Jia Lau, Ary A. Hoffmann.

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
