## [Decision Letter · Decision Letter 0]

3 Oct 2019

PONE-D-19-24952

Does membrane feeding compromise the quality of Aedes aegypti mosquitoes?

PLOS ONE

Dear Dr. Ross,

Thank you for submitting your manuscript to PLOS ONE. After careful consideration, we feel that it has merit but does not fully meet PLOS ONE’s publication criteria as it currently stands. Therefore, we invite you to submit a revised version of the manuscript that addresses the points raised during the review process.

We would appreciate receiving your revised manuscript by Nov 17 2019 11:59PM. To enhance the reproducibility of your results, we recommend that if applicable you deposit your laboratory protocols in protocols.io, where a protocol can be assigned its own identifier (DOI) such that it can be cited independently in the future. For instructions see: http://journals.plos.org/plosone/s/submission-guidelines#loc-laboratory-protocols

We look forward to receiving your revised manuscript.

Kind regards,

Nikos Vasilakis

Academic Editor

PLOS ONE

Journal Requirements:

1. In your Methods section, please provide additional location information of the mosquito collection sites, including geographic coordinates for the data set if available.

2. In your Methods section, please provide additional information regarding the permits you obtained for the work. Please ensure you have included the full name of the authority that approved the collection sites access and, if no permits were required, a brief statement explaining why.

Reviewers' comments:

Reviewer's Responses to Questions

**Comments to the Author**

1. Is the manuscript technically sound, and do the data support the conclusions?

Reviewer #1: Yes

Reviewer #2: Partly

2. Has the statistical analysis been performed appropriately and rigorously? 

Reviewer #1: I Don't Know

Reviewer #2: Yes

3. Have the authors made all data underlying the findings in their manuscript fully available?

Reviewer #1: Yes

Reviewer #2: Yes

4. Is the manuscript presented in an intelligible fashion and written in standard English?

Reviewer #1: Yes

Reviewer #2: Yes

5. Review Comments to the Author

Reviewer #1: This paper by Ross et al. examines the effect of repeated generational blood feeding by Aedes aegypti on artificial membrane devices to determine whether maintaining colonies in this manner that will later be released into nature will compromise their natural feeding success. This is a useful goal because several novel methods of mosquito control or manipulation to affect vector competence require the rearing and release of large numbers. The authors report that membrane feeding-selected mosquitoes exhibited higher feeding rates on membranes than mosquitoes maintained by feeding directly on a human, but this apparent adaptation did not have a major impact on their ability to feed on humans or their attraction to live hosts. The authors also discuss the possibility that inefficient blood feeding may result in colony population bottlenecks. Overall, the conclusions seem to be well supported by reasonably sound experimental design. One weakness is that the mosquitoes tested were all infected with a Wolbachia strain that is not a natural part of the Aedes aegypti microbiome. This may limit extrapolation of the results to other arbovirus control strategies not involving this bacterium. Another limitation is that these Ae. aegypti mosquitoes all fed to repletion, while natural human feeding generally involves multiple, partial blood meals. Also, the blood sources for the feeding success analyses were not the same individual, so interhuman differences could have affected the feeding results, and interhuman variance was not assessed. It is also unclear why at least some of the landing mosquitoes in the outdoor enclosure were not allowed to engorge, as this would have been a more natural indication of feeding success in the wild. Adult longevity would also be very useful to test, as this is a key factor in the vectorial capacity equation. All of these limitations should at least be mentioned in the discussion.

Minor comments:

1. I recommend a more complete description of membrane feeders in the introduction; a figure would be useful for this.

2. Line 142: Sugar-starved?

3. Line 145: Was the human blood anticoagulated (if so, what method) or defibrinated? Some contact should be provided on this method – is it the most common method of artificial bloodmeal preparation?

4. Line 160: Fitness is generally defined as reproductive success, while the results described in this and the subsequent paragraph include size, development time etc. that are not directly related to fitness.

5. Line 475: Since there is no direct evidence for “inbreeding depression,” this statement should be tempered. Are there estimates of population sizes associated with membrane feeding bottlenecks? It would also seem that programs designed to rear large numbers of mosquitoes for release would be unlikely to experience repeated bottlenecks after their establishment.

Reviewer #2: The study by Ross and co-workers examines if membrane feeding, a process commonly undertaken in lab-rearing of mosquitoes, compromises mosquitoes fitness. As the authors highlight, this is particularly relevant given that several novel vector control approaches are currently mass rearing mosquitoes for field release. As such, the work here is timely and will be of general interest to the vector biology community. The studies are well designed and the analysis appears appropriate. My main issues is the conclusions the authors come to in regards to their findings. This work found that there were some deleterious fitness effect resulting from membrane feeding, such as fecundity. This would important for releases which aim to spread elements (genes or bacteria) into a population. I appreciate the authors are likely being cautious in their language using terms such as “probably not substantially” but I feel it would be good to discuss how these fitness effect could influence releases and what strategies researchers could undertake to overcome any of these fitness limitations prior to release.

L24 – I find this conclusion is not supported by the previous statement which shows that membrane-adapted mosquitoes had compromised host seeking ability in semi-field cages. In particular, given how the paper is frame around investigating membrane feeding in the age of mosquito releases.

L26 – large is arbitrary. Could you please define large here?

L55. Transmit pathogens. Diseases cannot be transmitted. Disease is the symptom(s) that manifest in an infect host.

L493-4. Could this not be alleviated by introgression? If so, please add a qualifier to the statement.

L501. Is this related to the Wolbachia induced wobbly proboscis phenotype?

L510. This hypothesis is valid, but the authors did not find evidence to support it. However, these behaviours are deeply ingrained into the mosquitoes and I’m wondering if they simple did not find evidence of this as the experiment was not run for long enough? I’m not suggesting extending the life of the experiment, but perhaps this could be mentioned, or some evidence could be provided that the experiment was indeed conducted for a suitable number of generations.

L523. Repopulating lab colonies with field material is the obvious solution to overcoming any inbreeding effects. Should this be mentioned in the conclusions as a strategy to over comes these subtle effects?

6. PLOS authors have the option to publish the peer review history of their article (what does this mean?). If published, this will include your full peer review and any attached files.

Reviewer #1: No

Reviewer #2: No

---

## [Author Response · Author response to Decision Letter 0]

7 Oct 2019

Editor comments:

1. In your Methods section, please provide additional location information of the mosquito collection sites, including geographic coordinates for the data set if available.

We only have information on the suburb where mosquitoes were collected, not the locations of individual collection sites.

2. In your Methods section, please provide additional information regarding the permits you obtained for the work. Please ensure you have included the full name of the authority that approved the collection sites access and, if no permits were required, a brief statement explaining why.

We now include this statement in the methods section: “No permits were required for mosquito collections; verbal permission was obtained from each household before setting up traps.”

Reviewers' comments:

Reviewer #1: This paper by Ross et al. examines the effect of repeated generational blood feeding by Aedes aegypti on artificial membrane devices to determine whether maintaining colonies in this manner that will later be released into nature will compromise their natural feeding success. This is a useful goal because several novel methods of mosquito control or manipulation to affect vector competence require the rearing and release of large numbers. The authors report that membrane feeding-selected mosquitoes exhibited higher feeding rates on membranes than mosquitoes maintained by feeding directly on a human, but this apparent adaptation did not have a major impact on their ability to feed on humans or their attraction to live hosts. The authors also discuss the possibility that inefficient blood feeding may result in colony population bottlenecks. Overall, the conclusions seem to be well supported by reasonably sound experimental design. One weakness is that the mosquitoes tested were all infected with a Wolbachia strain that is not a natural part of the Aedes aegypti microbiome. This may limit extrapolation of the results to other arbovirus control strategies not involving this bacterium. Another limitation is that these Ae. aegypti mosquitoes all fed to repletion, while natural human feeding generally involves multiple, partial blood meals. Also, the blood sources for the feeding success analyses were not the same individual, so interhuman differences could have affected the feeding results, and interhuman variance was not assessed. It is also unclear why at least some of the landing mosquitoes in the outdoor enclosure were not allowed to engorge, as this would have been a more natural indication of feeding success in the wild. Adult longevity would also be very useful to test, as this is a key factor in the vectorial capacity equation. All of these limitations should at least be mentioned in the discussion.

Thanks for pointing out these limitations. Regarding the blood sources for the feeding success experiments, we used a single human volunteer and a single source of blood for membrane feeding in all experiments. We now clarify this in statements in the “blood meal weight and feeding duration” and “feeding proportion” sections in the methods.

We agree that letting females feed in the semi-field cage would have been a useful way to realistically measure feeding success, but this could not be done in the same experiment since we were specifically testing for host-seeking ability. We now mention that more realistic experiments including multiple /partial blood meals (and a longer duration of selection) may reveal effects not seen here. We also agree that longevity would be useful to test but it’s difficult to test this in a realistic way in the lab since lifespans are much longer.

Minor comments:

1. I recommend a more complete description of membrane feeders in the introduction; a figure would be useful for this.

We now include images of human arm feeding and the membrane feeding apparatus in Fig 1.

2. Line 142: Sugar-starved?

We now change “starved” to “sugar-starved” throughout the manuscript

3. Line 145: Was the human blood anticoagulated (if so, what method) or defibrinated? Some contact should be provided on this method – is it the most common method of artificial bloodmeal preparation?

No, the blood was not anticoagulated or defibrinated. There doesn’t seem to be any consensus in the literature but many studies use unaltered blood directly from the Red Cross or a commercial supplier.

4. Line 160: Fitness is generally defined as reproductive success, while the results described in this and the subsequent paragraph include size, development time etc. that are not directly related to fitness.

We now refer to these as life history experiments.

5. Line 475: Since there is no direct evidence for “inbreeding depression,” this statement should be tempered. Are there estimates of population sizes associated with membrane feeding bottlenecks? It would also seem that programs designed to rear large numbers of mosquitoes for release would be unlikely to experience repeated bottlenecks after their establishment.

We did not estimate effective population sizes directly in this study, but we believe that this is a likely explanation. In our previous study we maintained laboratory populations at different census sizes where we observed lower effective population sizes and reduced fitness in smaller populations (Ross et al. 2019 A comprehensive assessment of inbreeding and laboratory adaptation in Aedes aegypti mosquitoes). Effects in the membrane-selected populations are similar to those observed in populations kept at a low census size (100 individuals) or that passed through a bottleneck (isofemale lines). Due to the lack of adaptation observed here, repeated bottlenecks may have occurred during mass-rearing depending on the membrane feeding apparatus. However, this will be less of an issue if populations are very large in the first place. We now also state that these issues can be avoided through outcrossing to wild populations in the discussion.

Reviewer #2: The study by Ross and co-workers examines if membrane feeding, a process commonly undertaken in lab-rearing of mosquitoes, compromises mosquitoes fitness. As the authors highlight, this is particularly relevant given that several novel vector control approaches are currently mass rearing mosquitoes for field release. As such, the work here is timely and will be of general interest to the vector biology community. The studies are well designed and the analysis appears appropriate. My main issues is the conclusions the authors come to in regards to their findings. This work found that there were some deleterious fitness effect resulting from membrane feeding, such as fecundity. This would important for releases which aim to spread elements (genes or bacteria) into a population. I appreciate the authors are likely being cautious in their language using terms such as “probably not substantially” but I feel it would be good to discuss how these fitness effect could influence releases and what strategies researchers could undertake to overcome any of these fitness limitations prior to release.

We now discuss this issue in the concluding paragraph, acknowledging that there are some fitness costs but that outcrossing can be used to increase the fitness of mosquitoes prior to releases. 

L24 – I find this conclusion is not supported by the previous statement which shows that membrane-adapted mosquitoes had compromised host seeking ability in semi-field cages. In particular, given how the paper is frame around investigating membrane feeding in the age of mosquito releases.

Although host-seeking in the semi-field cage was compromised, we stand by our conclusion that field performance will not be compromised substantially, given that most other traits were unaffected and the effects we did observe may have been due to inbreeding rather than adaptation.

L26 – large is arbitrary. Could you please define large here?

We now define this as thousands of individuals.

L55. Transmit pathogens. Diseases cannot be transmitted. Disease is the symptom(s) that manifest in an infect host.

We now use pathogen or virus throughout the manuscript.

L493-4. Could this not be alleviated by introgression? If so, please add a qualifier to the statement.

We now add introgression as an example of how genetic background can be controlled.

L501. Is this related to the Wolbachia induced wobbly proboscis phenotype?

No, this is unrelated. We only tested relatively young females, while the bendy proboscis phenotype appears only in old (~>20 d old) females with the deleterious wMelPop Wolbachia infection.

L510. This hypothesis is valid, but the authors did not find evidence to support it. However, these behaviours are deeply ingrained into the mosquitoes and I’m wondering if they simple did not find evidence of this as the experiment was not run for long enough? I’m not suggesting extending the life of the experiment, but perhaps this could be mentioned, or some evidence could be provided that the experiment was indeed conducted for a suitable number of generations.

Shifts in these traits may indeed be very slow and it would be worth testing populations that have been maintained on membranes for a very long time in future studies. However, from the perspective of field releases, it is unlikely that mosquitoes will be maintained in the lab for more than a year without introgression / outcrossing. We now state this in the second last paragraph of the discussion.

L523. Repopulating lab colonies with field material is the obvious solution to overcoming any inbreeding effects. Should this be mentioned in the conclusions as a strategy to over comes these subtle effects?

We now state that outcrossing can be used to alleviate these issues in the final sentence.

---

## [Editor Report · Decision Letter 1]

10 Oct 2019

Does membrane feeding compromise the quality of Aedes aegypti mosquitoes?

PONE-D-19-24952R1

Dear Dr. Ross,

We are pleased to inform you that your manuscript has been judged scientifically suitable for publication and will be formally accepted for publication once it complies with all outstanding technical requirements.

With kind regards,

Nikos Vasilakis

Academic Editor

PLOS ONE
---

## [Editor Report · Acceptance letter]

14 Oct 2019

PONE-D-19-24952R1 

Does membrane feeding compromise the quality of *Aedes aegypti* mosquitoes? 

Dear Dr. Ross:

I am pleased to inform you that your manuscript has been deemed suitable for publication in PLOS ONE. Congratulations! Your manuscript is now with our production department. 

With kind regards,

on behalf of

Dr. Nikos Vasilakis 

Academic Editor

PLOS ONE